# Free Amino Nitrogen in Brewing

**Annie E. Hill *** and **Graham G. Stewart**

International Centre for Brewing & Distilling, Heriot-Watt University, Riccarton, Edinburgh EH14 4AS, Scotland; profggstewart@aol.com

*   Correspondence: a.hill@hw.ac.uk; Tel.: +44-1314513458

**Abstract:** The role of nitrogenous components in malt and wort during the production of beer has long been recognized. The concentration and range of wort amino acids impact on ethanolic fermentation by yeast and on the production of a range of flavour and aroma compounds in the final beer. This review summarizes research on Free Amino Nitrogen (FAN) within brewing, including various methods of analysis.

**Keywords:** brewing; fermentation; free amino nitrogen; wort; yeast

## 1. Introduction

The earliest written account of brewing dates from Mesopotamian times [1]. However, our understanding of the connection with yeast is relatively recent, starting with Leeuwenhoek's microscope observations in the 17th century followed by the work of Lavoisier, Gay-Lussac, Schwann and others during the 18th and 19th centuries. It was not until the late 19th century that Pasteur demonstrated that fermented beverages result from the action of living yeast's transformation of glucose (and other sugars) into ethanol [2–4]. Since then, our knowledge has expanded exponentially, particularly with the development of molecular biology techniques [5]. In this review, we cover the particular contribution that wort nitrogen components play in beer production during fermentation.

A number of terms are used to define wort nitrogenous components: Free Amino Nitrogen (FAN) is a measure of the nitrogen compounds that may be assimilated or metabolised by yeast during fermentation. These include individual wort alpha amino acids (with the exception of proline, which is not an $\alpha$-amino acid), ammonia and small peptides (di- and tri-peptides). Other terms for FAN are Primary Amino Nitrogen (PAN), Total Usable Nitrogen or Usable Nitrogen. Yeast Assimilable Nitrogen (YAN) is usually used in a discussion of wine must and is a measure of both free $\alpha$-amino acids and ammonia. For the purpose of this review, FAN will be used as the relevant term.

### 1.1. Sources of Nitrogen

The predominant raw material for brewing and distilling is malted barley, with wheat, maize, corn, rice, oats or sorghum often used and added as adjuncts. The total protein content, the predominant nitrogen-containing portion, of cereal seeds vary from about 10 to 15% of the grain dry weight [6]. Barley grains contain four distinct classes of protein: albumin, globulin, prolamin (hordein) and glutelin. The storage proteins, prolamin and glutelin, account for approximately 50% of the total protein in mature cereal grains and, with the exception of oats and rice, the major endosperm storage proteins of all cereal grains are prolamins [7]. Prolamins vary from 10 to 100 kDa in molecular mass and are generally relatively rich in the amino acid proline and amide nitrogen derived from glutamine (30–70% of the total), in addition to other specific amino acids such as histidine, glycine, methionine and phenylalanine. Generally, prolamins are deficient in lysine, threonine and tryptophan [6–8].

The prolamins of maize, sorghum and millet are rich in methionine, with maize being particularly low in tryptophan.

During malting, barley grain germination is initiated by the uptake of water (steeping). This promotes the development of enzymes required to modify starch reserves into fermentable sugars. It is at this stage that proteolytic enzymes are also activated. Proteolysis is important during malting because the soluble nitrogen pool required for enzyme synthesis is produced when proteolysis is optimal [9]. Optimal proteolysis results in the release of bound α-amylase enzymes, which are required for starch degradation.

Germinating barley grains contain a range of proteolytic enzymes including at least 40 different endoproteinases with high activities found in the aleurone layer and endosperm during germination [6,10]. The starchy endosperm contains approximately two-thirds of the grain's total reserve of proteins, and its internal pH during germination is 5.0–5.2. Carboxypeptidases are highly active at this pH because of their high concentration in the endosperm; it is likely that they have a central role in the mobilization of reserve proteins during germination [6]. High peptidase activities within the modified seed leaf (scutellum) of the grain suggests that some of the hydrolysis products are absorbed as peptides, and these are further hydrolysed to amino acids in this tissue [6,10]. Agu (2003) has determined that the nitrogen content of malting barley is linked to both enzyme production during malting and carbohydrates/soluble nitrogen present in the wort when the malted barley is subsequently mashed [9]. It is estimated that up to 70% of wort FAN is produced during malting [10,11]. In general, higher nitrogen barleys produce extracts that are rich in FAN [9]. In contrast, lower nitrogen barleys produce extracts that are rich in carbohydrates. Although nitrogen levels vary depending on the grain variety, the overall types of amino acids present are similar.

The use of sorghum in brewing has been keenly researched over the last 30 years [12–17]. As described above, FAN is largely derived from the breakdown of endosperm proteins during malting [18]. However, when sorghum is malted, much of the nitrogen in the kernel is transferred to the roots and shoots. Taylor (1983) demonstrated that, as with barley, prolamins are the major group of storage proteins; however, the FAN composition of malted grains considerably differs [19]. Malted barley is much richer in proline, whereas the two most important free amino acids in sorghum malt are asparagine and glutamine. In terms of yeast nutrition, sorghum malt has a higher percentage of readily assimilable amino acids than barley malt and other cereals such as wheat [20].

### 1.2. Wort Nitrogen

Following on from malting, the next stage in the brewing process is mashing, a process in which malt (and/or exogenous) enzymes break down (hydrolyse) cereal starches into fermentable sugars. As described above, the majority of wort FAN is produced during malting, however, endoproteinases are not destroyed during controlled kilning and, therefore, remain active during mashing [21,22]. The further extraction of soluble protein and FAN into wort is mainly dependent upon the mashing regime used. Mashing temperatures for optimal proteolytic enzyme activity are between 40 and 50 °C at an optimal pH of 3.8 [10]. Approximately 20–30% of the total wort FAN can be extracted during mashing [23].

The use of cereal adjuncts, such as unmalted barley, rice or wheat, can also impact on the extraction of soluble nitrogen during mashing. Using mixtures of unmalted barley and malt, an inhibitory effect on malt endopeptidases is observed resulting in a decrease of wort FAN, with no impact on the extraction of carbohydrates [24].

### 1.3. Amino Acid Uptake and Metabolism during Wort Fermentation

When yeast is pitched into wort, it is introduced into an extremely complex environment [5]. Wort consists mainly of fermentable sugars, including fructose, sucrose, glucose, maltose and maltotriose, and the remaining constituents are dextrins, nitrogenous materials, vitamins, ions, mineral salts, and trace elements [25]. Wort functions as both a growth medium to develop new yeast cells and a

fermentation medium for the yeast to produce ethanol, carbon dioxide and other metabolic products, many of which influence the flavour of the resulting beer [26]. During fermentation, brewing yeasts are required to adapt rapidly to this rich, concentrated environment, using the available nitrogen for the synthesis of cellular proteins and other cell compounds.

Approximately 10% of yeast dry weight is nitrogen. In wort, the main nitrogen source for the synthesis of proteins, nucleic acids and other nitrogenous cell components is the variety of amino acids, peptides, proteins, nucleic acids and their degradation products formed from the proteolysis of barley proteins described above. Brewer's wort contains 19 of the 20 essential amino acids, and it has been speculated that up to 400 dipeptides and over 8000 tripeptides may exist in malt wort as potential sources of nitrogen [27].

During the 1960s, Margaret Jones and John Pierce, working in the Guinness Research Laboratories, conducted studies on nitrogen metabolism during fermentation [28,29]. Their work has been substantiated by a number of more recent studies [23,30,31] showing that the absorption and utilization of exogenous nitrogenous wort compounds and their synthesis intracellularly are controlled by three main factors:

i.　　　The total wort concentration of assimilable nitrogen;
ii.　　　The concentration of individual nitrogenous compounds and their ratio;
iii.　　　The competitive inhibition of the uptake of these components (mainly amino acids) [28].

Jones and Pierce established a unique classification of amino acids according to their rates of consumption during typical ale brewing wort fermentations (Table 1) [28].

**Table 1.** Uptake of amino acids from wort. From Jones and Pierce (1964) [28].

| Group A | Group B | Group C | Group D |
|---|---|---|---|
| Fast absorption | Intermediate absorption | Slow absorption | Little or no absorption |
| Glutamate | | | |
| Aspartate | | Glycine | |
| Asparagine | Valine | Phenylalanine | |
| Glutamine | Methionine | Tyrosine | |
| Serine | Leucine | Tryptophan | Proline |
| Threonine | Isoleucine | Alanine | |
| Lysine | Histidine | Ammonia | |
| Arginine | | | |

Enari et al. (1964) suggested some changes to the groupings given in Table 1, and Lekkas et al. (2014) recommend that methionine, lysine and isoleucine uptake be moved to Group A from Group B, but the overall pattern of preferential uptake remains unchallenged (Figure 1) [23,24].

There are three ways by which yeast transports materials into the cell: diffusion, facilitated diffusion and active transport. Both diffusion and facilitated diffusion involve the movement of a substance across the plasma membrane from a high concentration to a lower concentration until equilibrium is reached. In facilitated diffusion, this movement is assisted by a protein but does not require energy. In active transport, movement across the plasma membrane is protein-mediated and generally involves energy. Using active transport, a substance can be accumulated against a concentration gradient, and it is by this method that amino acids are typically taken up by yeast. A range of amino acid transport mechanisms have been identified in *Saccharomyces cerevisiae*; a general amino acid permease (GAP) that transports several amino acids is found in the plasma membrane, in addition to 19 amino acid permeases that transport specific or a closely related group of amino acids [32].

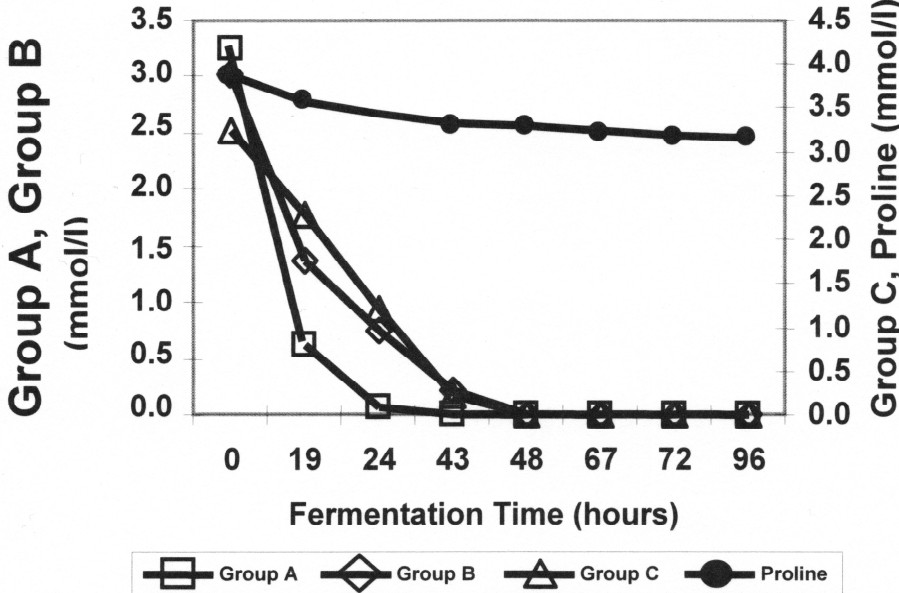

**Figure 1.** Amino acid adsorption patterns during wort fermentation. From Stewart, Hill and Lekkas (2013) [31].

The mechanisms by which yeasts sense and regulate the uptake of amino acids and other available nitrogen sources is complex. Magasanik and Kaiser (2002) have summarised nitrogen metabolism in *S. cerevisiae* as the result of three distinct elements:

i.    The enzymes responsible for the synthesis and interconversion of nitrogenous compounds;

ii.   The permeases for the uptake of nitrogenous compounds;

iii.  The transcription factors and membrane trafficking proteins that regulate the activity of the enzymes and permeases [33].

To prevent the uptake of non-preferred nitrogen sources at the start of fermentation, yeast uses a nitrogen catabolite repression mechanism [34]. The expression of nitrogen-regulated genes is activated by two transcription factors, and glutamine and glutamate serve as the intracellular signals to prevent this activation. In some cases, the enzymes responsible for the utilization of a non-preferred source of nitrogen are induced by the presence of the particular nitrogen source in the growth medium [34]. For example, the expression of the genes coding for the enzymes needed for the uptake (permeases) and utilization of arginine as a source of nitrogen is induced by the presence of arginine in the growth medium. The presence of a preferred source of nitrogen can lead to decreased activity of other permeases and, as a consequence, the induction of the specific nitrogen utilization pathways will be decreased [33].

GAP is inhibited by ammonium ions. As a result, it is active later in fermentation when the levels of ammonium ions decrease. Amino acid transport is also strongly inhibited by ethanol. When a significant amount of ethanol is produced during fermentation, the yeast membrane becomes permeable and allows influx of protons by diffusion. These excess protons in the cell are removed by a proton pump. However, to prevent overloading the pump's capacity to export protons, the cell shuts down $H^+$ ion-coupled amino acid transport, reducing the intake of both protons and amino acids [35].

Generally, yeast takes up amino acids early in fermentation when the ethanol concentration is relatively low and accumulates and stores them in vacuoles for later use when they are needed for metabolic activity [36]. This mechanism gives the yeast cell a competitive advantage because it depletes nutrients from the medium, depriving other organisms from obtaining them.

The metabolism of assimilated amino nitrogen is dependent on the phase of the fermentation and on the total quantity provided in the wort. Yeast can synthesize most of the amino acids needed to

build cellular proteins using ammonium ion and wort amino acids, with glutamate playing a vital role; glutamate can serve as an amino group donor in order to produce different amino acids. Yeast can also degrade amino acids to obtain ammonium ions. Ammonium ions and glutamate are used directly in biosynthesis [Figure 2] and are therefore preferred nitrogen sources for yeast growth [33,35]. Other preferred nitrogenous compounds include: glutamine, alanine, serine, threonine, aspartate, asparagine, arginine and urea [35].

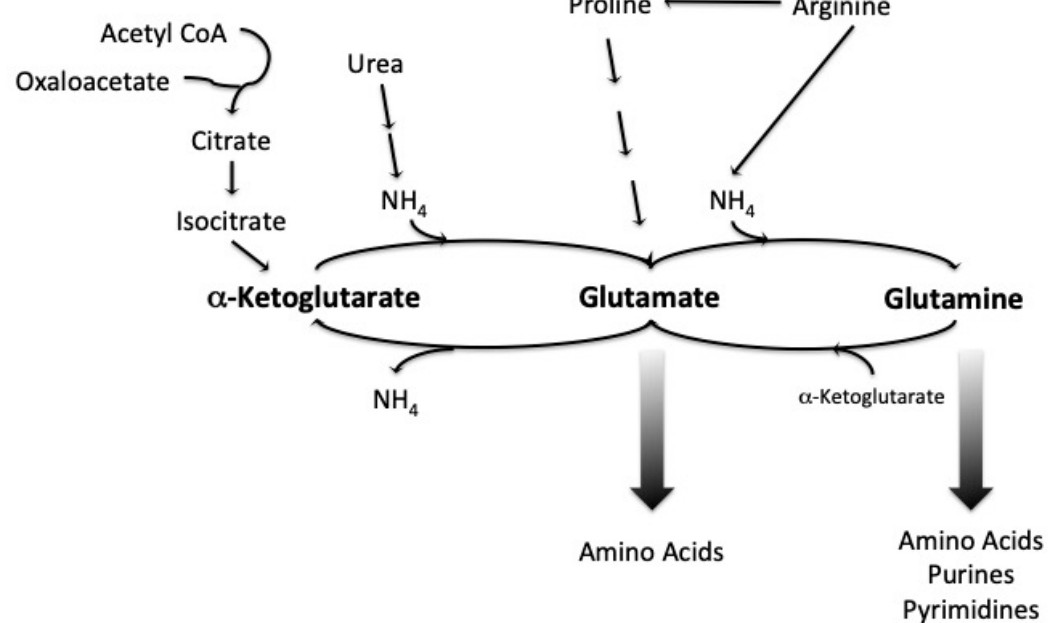

**Figure 2.** Central pathways for the utilization of a variety of nitrogen sources. Extracted from Magasanik and Kaiser (2002), with permission [33].

The non-preferred source of nitrogen used in most studies of nitrogen regulation is proline, which supports the classification of this amino acid in Group D (Table 1). The uptake of proline requires the presence of a mitochondrial oxidase; therefore, in the anaerobic conditions of wort fermentation, yeast cannot use proline as a nitrogen source. As a result, proline is usually still present in beer at 200–300 mg/L [37].

Although yeast can directly use a range of amino acids, the majority are involved in transamination reactions, and a significant proportion of the amino acid skeletons that are subsequently found in yeast proteins is derived from wort sugars. For this reason, the total amino content of wort is important in determining the extent of yeast growth, rather than the types of amino acid (although this is important for determining beer flavour, as discussed later) [38].

*1.4. Peptide Uptake*

Mass balance has determined that absorption of other nitrogenous components must accompany amino acid uptake [38]. Most yeast strains consume peptides no larger than tripeptides. Analysis of the uptake of fluorescent-labelled di- and tri-peptides has shown that they share the same transport system in *S. cerevisiae* [39]. No marked differences were found in the rates of peptide uptake by a wild-type strain and by a number of amino acid permease-deficient mutants, demonstrating that transport systems for peptides are separate to those for amino acids and they do not compete for uptake by the cell [39]. Studies of lager and ale yeast strains have subsequently shown that they can simultaneously use amino acids and small peptides as sources of assimilable nitrogen [37,40]. It is proposed that extracellular yeast proteolytic enzymes are responsible for the degradation of larger wort peptides into smaller molecules during fermentation to provide yeast cells with more available

assimilable nitrogen sources [40]. It is believed that only 40% of the total oligopeptides available in wort are used for nitrogen metabolic activity and that the rest may contribute to haze development (polypeptide–polyphenol complexes) or foam stability [41].

*1.5. Impact of Wort Nitrogen*

There has been much debate regarding the minimal FAN required to achieve satisfactory yeast growth and fermentation performance in normal gravity (10–12 °P) wort, and it is generally agreed to be around 130 mg FAN/L [41], with the minimum varying between 100 mg FAN/L and 140 mg FAN/L [42]. This is consistent across both brewing and wine fermentations. Below 100 mg FAN/L, yeast growth is nitrogen-dependent, and sub-optimal concentrations of available nitrogen are associated with lagging, incomplete fermentation and sulfide evolution [43]. In the stationary phase of yeast development, only low levels are required as a fermentation stimulant of the yeast, while higher levels are required during the growth phase.

Optimum FAN levels differ from fermentation to fermentation and also with yeast strain, wort sugar levels and type [30,31,44]. For wine fermentation, the optimal nitrogen concentration in the must is 190 mg FAN/L [45], with similar levels (200–250 mg FAN/L) regarded as optimal for standard gravity brewery fermentations. There are differences between lager and ale yeast strains with respect to wort-assimilable nitrogen uptake characteristics. However, with all brewing strains, the amount of wort FAN content required by yeast under normal brewery fermentation is directly proportional to yeast growth [5]. For rapid attenuation of high-gravity wort (16 °P), increased levels of FAN are required [30]. This results in more rapid initial fermentation of such worts as long as the dissolved oxygen level is increased on a pro rata basis at the beginning of fermentation [43].

Fermentation results indicate that wort FAN correlates well with at least three fermentation performance indicators. First, high initial FAN content allows a more efficient reduction of the wort gravity; secondly, the pH decrease during fermentation is proportional to the amount of FAN utilized; thirdly, the wort FAN content is a useful index towards the formation of total vicinal diketones, esters and higher alcohols during the later stages of fermentation [5].

The impact of FAN on the formation of flavour and aroma compounds during fermentation has been widely studied, particularly during wine production. Both the initial wort or must FAN content and the amino acid and ammonium ion equilibrium in the medium impact on the formation of esters, aldehydes, vicinal diketones, higher alcohols and acids, as well as sulfur compounds [26]. Even small differences in wort composition can exert significant effects on the flavour of the resulting beer [26]. Nitrogen concentration has been shown to impact on at least 23 compounds in wine: branched-chain fatty acids and their esters are associated with low nitrogen concentrations, whereas medium-chain fatty esters and acetic acid are associated with high nitrogen concentrations [45–48]. In brewing, excess FAN levels are associated with the production of off-flavours such as diacetyl and higher alcohols including isoamyl alcohol, propanol and isobutanol (details later). Conversely, cells starved of FAN (or oxygen) produce very low levels of esters.

A closer examination of flavour formation reveals an impact of wort amino acid composition. Subdivision of the classes detailed in Table 1 may be made on the basis of their "essential" nature (Table 2).

The initial concentration of Class 1 amino acids is considered relatively unimportant, since they may be incorporated directly from the wort when available or synthesised from sugar metabolism and transamination in later fermentation. Deficiencies in Class 2 and Class 3 amino acids, however, have considerable effects on beer quality. In the later stages of fermentation, when the supply of exogenous amino acids is exhausted, the keto-acid moiety of Class 2 amino acids must be synthesised solely from sugars. This process of keto-acid formation results in carbonyl by-products, such as diacetyl, which impart off-flavours to the beer. For Class 3 amino acids, the contribution made by the sugar synthetic route is small, and the yeast is dependent on an adequate exogenous supply. Therefore, a deficiency in Class 3 amino acids results in major perturbations in nitrogen metabolism, with significant repercussions on beer flavour [29].

**Table 2.** Classification of amino acids on the basis of their essential nature. From Pierce (1987) [29].

| Class 1 | Class 2 | Class 3 |
|---------|---------|---------|
| Glutamate | | |
| Aspartate | Valine | |
| Asparagine | Isoleucine | Leucine |
| Serine | Phenylalanine | Histidine |
| Threonine | Glycine | Lysine |
| Methionine | Tyrosine | Arginine |
| Proline | | |

*1.6. Erlich Pathway*

It has already been discussed that brewer's yeast strains absorb wort's spectrum of 19 amino acids together with a number of small peptides [40]. The amino group is removed so that it can be incorporated into other structures. What remains from the amino acids are $\alpha$-keto acids which enter into an irreversible chain reaction that will ultimately form higher alcohols (Figure 3). This is the Erlich pathway, which has led to an investigation of the relationship between isoamyl alcohol with leucine (Figure 3).

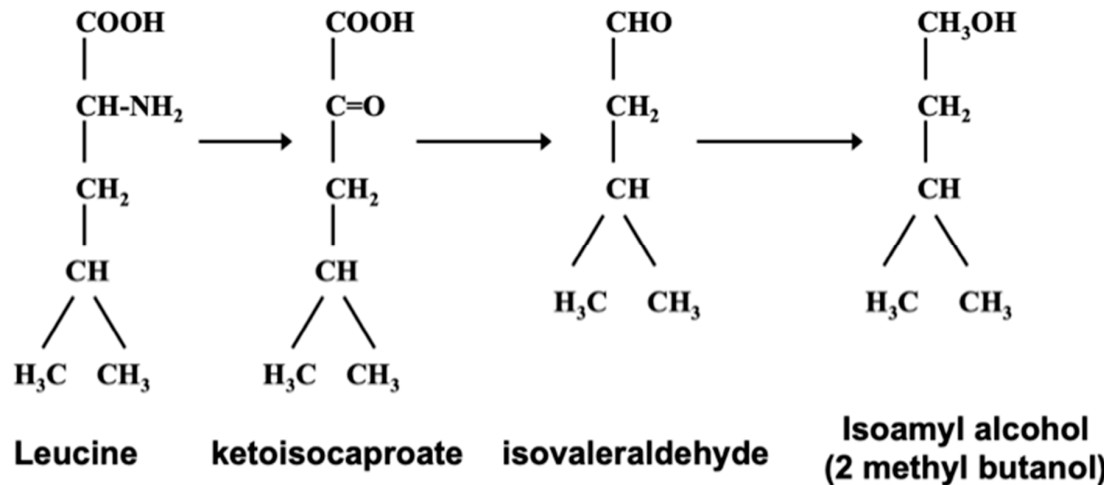

**Figure 3.** Ehrlich pathway, relationship between higher (fusel) alcohols and amino acids.

Erlich also proposed that amino acids were enzymatically hydrolyzed to form the corresponding higher alcohol, along with ammonia and carbon dioxide. As ammonia was not detected in the reaction medium, it has been assumed that it was incorporated directly into proteins [49]. Subsequently, intermediate steps to this pathway have been proposed that complete this metabolic pathway. In addition, higher alcohols are also formed during the upstream (anabolic pathway) biosynthesis of amino acids.

It is apparent that the amino nitrogen composition of wort has far-reaching effects upon fermentation performance and beer flavour. Other implications of excess FAN in beer include issues with beer haze and microbiological stability. Where malt is used as the principal source of fermentable extract, the quantity and composition of amino acids are such that these problems are, on the whole, not encountered. However, care must be exercised when using adjuncts, most of which are deficient in amino nitrogen.

## 2. Measuring Wort Nitrogen

We have found that optimal industrial yeast fermentation performance and final beer quality are influenced by the complexity of the nitrogen source. The traditional analytical method for quantitative determination of nitrogen contained in organic substances and ammonia is the Kjeldahl method or Kjeldahl digestion developed by Johan Kjeldahl in 1883. This method involves heating with sulphuric acid, followed by distillation with sodium hydroxide and then titration, and is internationally recognized for estimating protein content in foods [50].

A more rapid and automatable technique than Kjeldahl's is the Dumas method. The Dumas method consists of combusting a sample of known mass in a chamber at a high temperature range of 800–900 °C in the presence of oxygen [51]. This leads to the release of carbon dioxide, water and nitrogen. The gases are then passed over columns that absorb the carbon dioxide and water. A column containing a thermal conductivity detector at the end is then used to separate the nitrogen from any residual carbon dioxide and water, and the remaining nitrogen content is measured against a standard of known nitrogen concentration.

Both the Kjeldahl and the Dumas methods require the conversion of nitrogen in the sample to crude protein content using conversion factors, and neither give the true protein measurement. Therefore, more accurate techniques now examine the amino acid content. Amino acids lack natural strong chromophore or fluorophore groups, making them difficult to detect using photometry or fluorometry. However, primary amino groups react with ninhydrin to form a purple dye (Ruhemann's purple), and other amino acid groups also react with ninhydrin to form various chromophores that may be analysed. These reactions form the basis of the Ninhydrin or Kaiser test for the determination of free amino acids [52–56]. The sensitivity of the Ninhydrin test is generally sufficient for most purposes, and high-throughput microwell plates are available, including kits such as Spectrostar® Nano and QuantiChrom™ [57–59] For the quantification of specific amino acids, separation may be achieved using liquid chromatography and revealed using the ninhydrin reaction, followed by absorbance measurements at 570 nm (except for proline, which is read at 440 nm). An alternative is to use gradient elution high-performance liquid chromatography (HPLC) with a fluorescence detector, using dansyl chloride as the fluorogenic reagent [60–62].

More recently, techniques using o-phthaldialdehyde (OPA) have been developed, including an automated method that has been approved by the European Brewery Convention [63]. This method, called alpha amino nitrogen by OPA (NOPA), determines the amino acid content in beer using a photometric measurement of OPA and N-acetyl cysteine (NAC). The results correlate well with the Kjedhal and ninhydrin methods as a measure of FAN and do not require additional ammonia measurement.

For a more complete understanding of the nitrogeneous components in wort or beer, the quantification of ammonia, ammonium ions and oligopeptides should be included. A range of colourimetric methods are available for the determination of both ammonia and ammonium ions, including spectrophotometric assay kits such as Enzytec™.

Oligopeptides (di-and tri-peptides) may be determined using an HPLC method with initial hydrolysis to break the peptide bonds between constituent amino acids [40]. Acid hydrolysis destroys tryptophan, consequently an alkaline hydrolysis step is typically included, and unhydrolysed samples are also analysed to calculate the concentration of individual non-peptide-bonded amino acids. Oligopeptide levels are estimated by subtracting the values obtained with the unhydrolysed samples from those of the hydrolysed samples.

## 3. Summary

Our knowledge of the roles of nitrogenous components of malt and wort to meet yeast requirements has substantially increased over the years. Nevertheless, the optimization of the nitrogen content of wort is a very complex issue owing to the large number of nitrogen compounds found in the malt [5]. FAN measurement on its own is a 'blunt instrument' but, if used with other measurements, such as pH, bitterness, wort sugar and gravity, alcohol content and microbiology screening, it can

provide the brewer with valuable insight into the quality and consistency of their raw materials, process and product.

**Author Contributions:** Original draft preparation was carried out by A.E.H. with writing-review and editing by G.G.S.

**Funding:** This research received no external funding.

**Acknowledgments:** The authors are grateful to Anne Anstruther for her assistance in the preparation of this manuscript.

**Conflicts of Interest:** The authors declare no conflict of interest.

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
