# Peer review of "Free Amino Nitrogen in Brewing"

_fermentation, doi:10.3390/fermentation5010022_

Round 1
Reviewer 1 Report
Overall this is a well-written review of the role of FAN in brewing. The material seems thorough and does not require revision, however there are a number of grammar, formatting, and citation issues which should be addressed.
Line 22 - end sentence after "centuries", start new sentence after.
Lines 39-41 - statements about storage proteins need citations/references.
Line 41 - the proper unit for protein size is kDa, not k.
Lines 55, 69 - from how this is written, it seems that references 25, 29, and 38 should be in both.
Line 105 - indicates that more recent studies have been completed, but only the single, older reference is cited. Newer ones should be included.
Fix spacing in and around table 1.
Line 147 - change "activity of the permeases" to "activity of other permeases" to avoid confusion.
Line 159 - information in this paragraph needs citation.
Line 167 - Statement needs citation.
Line 248 - change "their own structures" to "other structures".
Line 259 - Statements need citations.
Line 272 - statements need citations.
Line 279 - statements need citations.
The last paragraph seems to be a summary. Review would benefit from a section 3 titled "Summary" or "Conclusions" with a somewhat more comprehensive summary/concluding statements.
Line 364 - Genus/species should be italicized.
Formatting is not compliant with journal instructions. Citations should be ordered by appearance in the text, not alphabetically in the reference section. In-text citations should be in square brackets, not superscripts.
Table titles should be above the table, not below.
Figure title/legends should be below the figure, not above.
Author Response
Reviewer 1
Overall this is a well-written review of the role of FAN in brewing. The material seems thorough and does not require revision, however there are a number of grammar, formatting, and citation issues which should be addressed.
Line 22 - end sentence after "centuries", start new sentence after.
Updated.
Lines 39-41 - statements about storage proteins need citations/references.
Shewry and Halford reference added.
Line 41 - the proper unit for protein size is kDa, not k.
Updated.
Lines 55, 69 - from how this is written, it seems that references 25, 29, and 38 should be in both.
Ref 29 added to Line 62.
Line 105 - indicates that more recent studies have been completed, but only the single, older reference is cited. Newer ones should be included.
References added.
Fix spacing in and around table 1.
Updated.
Line 147 - change "activity of the permeases" to "activity of other permeases" to avoid confusion.
Updated.
Line 159 - information in this paragraph needs citation.
Garrett, 2008
Line 167 - Statement needs citation.
Dharmadhikari, 2001
Line 248 - change "their own structures" to "other structures".
Updated.
Line 259 - Statements need citations.
Hazelwood et al., 2008
Line 272 - statements need citations.
Kirk, 1950
Line 279 - statements need citations.
Ebeling, 1968
The last paragraph seems to be a summary. Review would benefit from a section 3 titled "Summary" or "Conclusions" with a somewhat more comprehensive summary/concluding statements.
Separate section added.
Line 364 - Genus/species should be italicized.
Updated.
Formatting is not compliant with journal instructions. Citations should be ordered by appearance in the text, not alphabetically in the reference section. In-text citations should be in square brackets, not superscripts.
References renumbered.
Table titles should be above the table, not below.
Updated.
Figure title/legends should be below the figure, not above.
Updated.
Reviewer 2 Report
Authors of this manuscript have a broad experience and in-depth knowledge of the brewing and malting process.
The manuscript is very well written and well organized.
Only minor spell check is required, in particular:
L41: 10 to 100 kDa
L283: fluorophore
L284: fluorometry
L290: citation must be formatted
L298: citation must be formatted
Tables: title should appear before the table
Figures: caption should appear below the figure
Author Response
Reviewer 2
Authors of this manuscript have a broad experience and in-depth knowledge of the brewing and malting process.
The manuscript is very well written and well organized.
Only minor spell check is required, in particular:
L41: 10 to 100 kDa
Updated.
L283: fluorophore
Updated.
L284: fluorometry
Updated.
L290: citation must be formatted
Updated.
L298: citation must be formatted
Updated.
Tables: title should appear before the table
Updated.
Figures: caption should appear below the figure
Updated.